

# An optimal observable for color singlet identification

**Andy Buckley**[1★], **Giuseppe Callea**[1], **Andrew J. Larkoski**[2] **and Simone Marzani**[3]

**1** School of Physics & Astronomy, University of Glasgow, Glasgow G12 8QQ, Scotland, UK
**2** Physics Department, Reed College, Portland, OR 97202, USA
**3** Dipartimento di Fisica, Università di Genova and INFN, Sezione di Genova,
Via Dodecaneso 33, 16146, Italy

★ andy.buckley@cern.ch

## Abstract

We present a novel approach to construct a color singlet tagger, i.e. an observable that is able to discriminate the decay of a color-singlet into two jets from a two-jet background in a different color configuration. We do this by explicitly taking the ratio of the corresponding leading-order matrix elements in the soft limit, thus obtaining an observable that is provably optimal within our approximation. We call this observable the "jet color ring" and we compare its performance to other color-sensitive observables such as jet pull and dipolarity. We also assess the performance of the jet color ring in simulations by applying it to the case of the hadronic decays of a boosted Higgs boson and of an electroweak boson.



## 1 Introduction

The quantum numbers of particles produced at short distances in high-energy collisions are encoded in the radiation pattern of the particles observed in experiment. Mass is perhaps

the most familiar and most straightforward quantum number to observe in collider experiment, but observables have been constructed that are sensitive to a number of intrinsic particle properties. In particular, several observables have been designed to be sensitive to the representation under the SU(3) color of quantum chromodynamics (QCD) of pairs of jets that are detected close in angle to one another in experiment [1–5]. Of these color-sensitive observables, the most widely-studied is jet pull [1], which has been measured at the Tevatron by the DØ collaboration and at the Large Hadron Collider (LHC) by ATLAS [6–8], employed in searches by CMS [9,10], and calculated for various jet configurations [11–13].

In this paper, we introduce a new observable sensitive to the color representation of pairs of jets that is provably optimal given reasonable assumptions. We focus primarily on two signal processes that feature the decay of a color singlet, namely the decay of the Higgs bosons into (bottom-quark) jets and the decay of an electroweak boson into light quark jets.

Direct observation of the hadronic decays of the Higgs boson is a central goal of the LHC. For boosted $H \rightarrow b\bar{b}$ decays, the main background is from $g \rightarrow b\bar{b}$ splittings in QCD. This problem is significantly interesting because $H \rightarrow b\bar{b}$, though the dominant decay mode of the Higgs, was only recently observed by ATLAS and CMS [14,15]. Further, once mass is selected for, the kinematics of the bottom quarks in $H \rightarrow b\bar{b}$ and $g \rightarrow b\bar{b}$ are very similar because there is no soft singularity for a gluon splitting to quarks. Jet pull has been employed by experiments for this purpose, even featuring in the analysis of $H \rightarrow b\bar{b}$ decay in the discovery of the Higgs boson by the CMS experiment [16]. However, it has also been observed that in practice, the discrimination power of pull is not as strong as one might have expected [17–19], so it is desirable to construct a new observable that is both simply motivated and performant. Furthermore, while the Higgs decay into light jets will likely remain unreachable for the LHC, with results from the high-luminosity run only able to set loose bounds [20, 21], this decay channel could become the target for future electron-positron colliders.

The situation is less critical in the case of $W$ and $Z$ boson decaying into two jets. In this case the dominant background to $V \rightarrow q\bar{q}$ is constituted by QCD splittings that feature a different kinematic behavior because of the soft singularities related to the emissions of low-energy gluons. Nevertheless, we find it interesting to investigate how a tagger built exclusively on color information performs in this situation. Furthermore, this study can be formative for future studies of boosted electroweak bosons decaying into heavy flavors, which shares some similarities with the aforementioned Higgs decay.

Our approach to this problem will be to construct a new color-sensitive observable which we call the jet color ring. It is formed from the likelihood ratio of background to signal matrix elements, computed at leading-order in the perturbation theory of QCD and in the high energy or large Lorentz boost limit. While many observables on jets are motivated from physical considerations, this is the first case, to the best of our knowledge, in which the observable is simply defined from the ratio of matrix elements.[1]

The motivation for this variable is further enhanced by the experimental availability of small-$R$ jets constructed from charged-particle tracks, which can be used in conjunction with flavor-tagging techniques to provide a high-resolution estimate of the orientation and size of the radiating dipole within a jet. This information is not currently exploited by standard resonance taggers.

We compare the jet color ring to pull and to another color-sensitive observable called dipolarity [2], and explicitly demonstrate that pull and dipolarity are technically sub-optimal for this color discrimination problem, given our assumptions. We show that good discrimination power of the jet color ring is observed in simulation for the $H \rightarrow b\bar{b}$ process, but that this per-

---

[1]Our approach shares some similarities with the framework of shower deconstruction [22–25], which also considers ratios of matrix elements to approximate the likelihood. However, our target here is to exploit this approach to construct a simple and intuitive observable.

formance degrades significantly without the restriction to $b$-tagged jet constituents. Further, we compare to the $D_2$ tagging observable [26] and note that its performance is better than our analysis would expect, suggesting future directions of study incorporating higher-order QCD effects, and that a hybrid tagger using both the color ring and $D_2$ observables can outperform both individually.

## 2 Observable Derivation

We consider the decay of a boosted massive color singlet into two jets. At leading order (LO) in QCD perturbation theory this process is described by the decay of the massive particles into two massless partons. Each parton could be a (anti)quark, i.e. an object that transforms according to the (anti)fundamental representation of the color group SU(3), or a gluon, i.e. an object that transforms in the adjoint representation. By color conservation, the two partons recoil against the rest of the event, which has net color that is opposite to that of the combined boosted parton system.

The matrix element square describing the decay of a boosted color-singlet state into two hard partons plus a single low-energy gluon is

$$|\mathcal{M}_S|^2 = -\mathbf{T}_\alpha \cdot \mathbf{T}_\beta \frac{n_a \cdot n_b}{(n_a \cdot k)(n_b \cdot k)} \,. \tag{1}$$

Note that we have suppressed dependence on the strong coupling $\alpha_s$ as that will not be relevant for constructing the observable. In the above equation, $k$ is the momentum of the soft gluon and $n_a$ and $n_b$ are light-like vectors pointing in the direction of the outgoing jets. The color operator $\mathbf{T}_i$ accounts for the changes in color space due to the emission of a soft gluon off a hard parton, that can be either a quark ($i = q$) or gluon ($i = g$) [27–33]. Note that, following Ref. [34] we have kept separate indices to indicate parton kinematics $(a, b)$ and color $(\alpha, \beta)$. We can think of parton $a$, for instance, as the one which is left-moving in the decaying particle center of mass frame. This is not necessary here, but it will become useful when considering the background process.

Color conservation requires that

$$\mathbf{T}_\alpha + \mathbf{T}_\beta = 0 \,, \tag{2}$$

which implies that $\beta = \overline{\alpha}$, the anti-color of $\alpha$, and that, consequently, the color matrix product is

$$-\mathbf{T}_\alpha \cdot \mathbf{T}_\beta = \mathbf{T}_\alpha^2 = C_S \mathbf{1} \,, \tag{3}$$

an SU(3) quadratic Casimir and $\mathbf{1}$ is the identity matrix in color space. For instance, if we were considering a color singlet decaying into quarks such as $H \to b\bar{b}$, then $C_S = C_F$, i.e. the Casimir of the fundamental representation, as appropriate for final-state quark jets. We could also consider a color singlet decaying into gluons and in such case we would have $C_S = C_A$, i.e. the Casimir of the adjoint representation, as appropriate for final-state gluon jets.

The matrix element for soft gluon emission in the background process has a richer structure. Here, we will focus on the high-boost limit in which the final-state jets are relatively collimated. In particular, assuming that the final-state partons that seed the jets we are triggering on are at smaller angle to each other than any other colored direction in the event, we can factor the complete matrix element square for the background process into two pieces. One, a contribution describing the dipole formed by the two quasi-collinear partons plus soft radiation, which we refer henceforth as the background matrix element. Two, as a piece that includes both initial-state contributions such as parton distribution functions, as well as eventual extra jets. The latter contributes with an overall factor which is not relevant, within our

approximation, for the construction of the observable. The former can be approximated as having three relevant directions in which color flows.

With this assumption, the background matrix element for single soft gluon emission is:

$$|\mathcal{M_B}|^2 = \sum_{\alpha,\beta}\left[ -\mathbf{T}_\alpha\cdot\mathbf{T}_\beta\frac{n_a\cdot n_b}{(n_a\cdot k)(n_b\cdot k)} \right.$$
$$\left. -\mathbf{T}_\alpha\cdot\mathbf{T}_\gamma\frac{n_a\cdot\bar{n}}{(n_a\cdot k)(\bar{n}\cdot k)} -\mathbf{T}_\beta\cdot\mathbf{T}_\gamma\frac{n_b\cdot\bar{n}}{(n_b\cdot k)(\bar{n}\cdot k)} \right]. \tag{4}$$

The vector $\bar{n}$ is light-like and points in the direction opposite to the two-jet system and the sum over the color indices $\alpha,\beta$ run over all possibilities that contribute to the process of interest. The color associated to the $\bar{n}$ direction, $\gamma$, is fixed by color conservation:

$$\mathbf{T}_\alpha + \mathbf{T}_\beta + \mathbf{T}_\gamma = 0. \tag{5}$$

We can therefore eliminate $\mathbf{T}_\gamma$ in the background matrix element:

$$|\mathcal{M_B}|^2 = \sum_{\alpha,\beta}\left[ -\mathbf{T}_\alpha\cdot\mathbf{T}_\beta\frac{n_a\cdot n_b}{(n_a\cdot k)(n_b\cdot k)} \right. \tag{6}$$
$$+\left(\mathbf{T}_\alpha^2+\mathbf{T}_\alpha\cdot\mathbf{T}_\beta\right)\frac{n_a\cdot\bar{n}}{(n_a\cdot k)(\bar{n}\cdot k)}$$
$$\left. +\left(\mathbf{T}_\beta^2+\mathbf{T}_\alpha\cdot\mathbf{T}_\beta\right)\frac{n_b\cdot\bar{n}}{(n_b\cdot k)(\bar{n}\cdot k)} \right]$$
$$=\sum_{\alpha,\beta}\left[ -\mathbf{T}_\alpha\cdot\mathbf{T}_\beta\frac{n_a\cdot n_b}{(n_a\cdot k)(n_b\cdot k)} \right.$$
$$\left. +\left(\mathbf{T}_\alpha^2+\mathbf{T}_\alpha\cdot\mathbf{T}_\beta\right)\left(\frac{n_a\cdot\bar{n}}{(n_a\cdot k)(\bar{n}\cdot k)}+\frac{n_b\cdot\bar{n}}{(n_b\cdot k)(\bar{n}\cdot k)}\right) \right],$$

where we have exploited the fact that our analysis is blind to both the color and the flavor of the final-state partons. Note that also in this case, the color matrix products in Eq. (4) can be diagonalized

$$-\mathbf{T}_\alpha\cdot\mathbf{T}_\beta = \frac{1}{2}\left[\mathbf{T}_\alpha^2+\mathbf{T}_\beta^2-\mathbf{T}_\gamma^2\right], \tag{7}$$

which follows from rearranging and squaring Eq. (5). The background matrix element squared can be written in terms of two effective color factors $C_\mathcal{B}$ and $\widetilde{C}_\mathcal{B}$:

$$|\mathcal{M_B}|^2 = C_\mathcal{B}\frac{n_a\cdot n_b}{(n_a\cdot k)(n_b\cdot k)} + \widetilde{C}_\mathcal{B}\left(\frac{n_a\cdot\bar{n}}{(n_a\cdot k)(\bar{n}\cdot k)}+\frac{n_b\cdot\bar{n}}{(n_b\cdot k)(\bar{n}\cdot k)}\right). \tag{8}$$

Before continuing our discussion, let us consider a few examples. If we are interested $g\to b\bar{b}$ as the background process to $H\to b\bar{b}$, we then have $C_\mathcal{S}=C_F$, $C_\mathcal{B}=C_F-C_A/2$, and $\widetilde{C}_\mathcal{B}=C_A/2$. The color flow for the background process is depicted in Fig. 1 while the color flow for the signal is depicted in Fig. 2. On the other hand, if we are considering as signal process $H\to gg$ (or $V\to q\bar{q}$), with background all $1\to 2$ QCD splittings, we then have $C_\mathcal{S}=C_A(C_F)$. $C_\mathcal{B}$, on the other hand, will be some linear combination of $C_A$ and $C_F$, weighted by parton distribution functions that set the relative fractions of the flavors of QCD jets that compose the background.

By the Neyman-Pearson lemma [35], the optimal discriminant observable for distinguishing the momentum dependence of these color configurations is monotonic in their likelihood

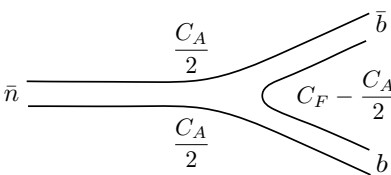

Figure 1: Illustration of the direction of color flow for the octet configuration of bottom ($b$) and anti-bottom ($\bar{b}$) quarks, in the collinear limit. All other colored particles in the event are present in the $\bar{n}$ direction, and the three pairs of color matrix products are listed.

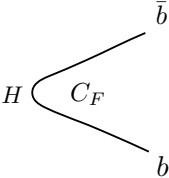

Figure 2: Illustration of the direction of color flow for the singlet configuration of bottom ($b$) and anti-bottom ($\bar{b}$) quarks from Higgs boson decay. The one non-trivial color matrix product is listed.

ratio. Taking the ratio of these matrix elements, we have

$$\frac{|\mathcal{M}_{\mathcal{B}}|^2}{|\mathcal{M}_{\mathcal{S}}|^2} = \frac{C_{\mathcal{B}}}{C_{\mathcal{S}}} + \frac{\widetilde{C}_{\mathcal{B}}}{C_{\mathcal{S}}}\left(\frac{(n_a \cdot \bar{n})(n_b \cdot k)}{(n_a \cdot n_b)(\bar{n} \cdot k)} + \frac{(n_b \cdot \bar{n})(n_a \cdot k)}{(n_a \cdot n_b)(\bar{n} \cdot k)}\right). \tag{9}$$

Because we only care about a function monotonic in this ratio, we can ignore the constant color factors for determining a discrimination observable. That is, we only consider

$$\frac{|\mathcal{M}_{\mathcal{B}}|^2}{|\mathcal{M}_{\mathcal{S}}|^2} \simeq \frac{(n_a \cdot \bar{n})(n_b \cdot k)}{(n_a \cdot n_b)(\bar{n} \cdot k)} + \frac{(n_b \cdot \bar{n})(n_a \cdot k)}{(n_a \cdot n_b)(\bar{n} \cdot k)}, \tag{10}$$

where $\simeq$ means up to a monotonic function.

To go further, we exploit the kinematics of the dipole configuration and the collinear limit. In the limit in which the final-state partons are collinear, to leading power in their splitting angle $\bar{n} \cdot k$ is just twice the energy of soft gluon $k$. Additionally, the dot products of light-like vectors is

$$\bar{n} \cdot n_a \simeq \bar{n} \cdot n_b \simeq 2, \tag{11}$$

again to leading power in the splitting angle of final-state hard partons. With these assumptions, we can further reduce the discrimination observable to

$$\frac{|\mathcal{M}_{\mathcal{B}}|^2}{|\mathcal{M}_{\mathcal{S}}|^2} \simeq \frac{1 - \cos\theta_{ak} + 1 - \cos\theta_{bk}}{1 - \cos\theta_{ab}}, \tag{12}$$

where $\theta_{ak}$ ($\theta_{bk}$) is the angle between the soft gluon and each of the final-state hard partons, and $\theta_{ab}$ is the angle between them.

Finally, we Taylor-expand the cosine factors in the collinear limit, and take our discrimination observable to be the ratio of angles:

$$\mathcal{O} = \frac{\theta_{ak}^2 + \theta_{bk}^2}{\theta_{ab}^2}. \tag{13}$$

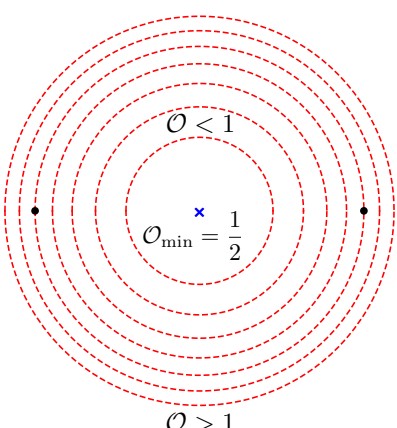

Figure 3: Illustration of the geometry selected for by the observable $\mathcal{O}$. The location of the final-state hard partons are denoted by the black dots, and the $\mathcal{O} = 1$ contour passes through the black dots. The region inside the $\mathcal{O} = 1$ circle has observable value less than 1 and the region outside is greater than 1. The point at the center of the circle, directly between the bottom quarks, is where the observable takes its minimum value, as labeled.

We refer to this observable as the *jet color ring*. Note that if we were to interpret Eq. (13) as the leading-order expression of a jet (or event) shape, we would conclude that jet color ring is not infrared and collinear (IRC) safe, as it is not weighted by the energy of the soft gluon emission $k$. This, in turn, maybe interpreted as an inability to theoretically predict its distribution on signal or background events. Such jet shape may be Sudakov safe [36–38], although it is not clear what its natural safe companion observable would be [2] However, we view Eq. (13) in a more positive light by interpreting the gluon with momentum $k$ to be the leading-order approximation of a well-defined subjet in the larger jet. We will come to the precise particle-level definition of the the jet color ring in terms of subjet kinematics in Sec. 3 when we test its discrimination power in simulation.

The form of the sum and ratio of squares of angles in Eq. (13) has a nice geometric interpretation. The observable $\mathcal{O}$ can take values greater or less than 1, depending on the location of emission $k$. An illustration of the bottom quark dipole configuration including several contours to guide the eye is shown in Fig. 3. All contours are circles, and $\mathcal{O} = 1$ for the circle with diameter equal to $\theta_{ab}$ with two antipodal points corresponding to the bottom and anti-bottom quark directions. $\mathcal{O}$ is less than 1 inside this circle, taking the minimum value at its center for which

$$\mathcal{O}_{\min} = 1/2 \,. \tag{14}$$

Emissions from a color-singlet dipole dominantly lie inside, while emissions from a color-octet dipole dominantly lie outside the $\mathcal{O} = 1$ circle.

## 2.1 Relationship to the Pull Angle

For a single emission off of the dipole, the pull angle $\phi_p$ [1] can be defined and compared to the observable $\mathcal{O}$ defined from the likelihood ratio. The pull angle is the azimuthal angle of the soft gluon emission about the hard parton closest to the emission, with respect to the line

---

[2]In a recent study [39], a generalized defintion of IRC safety (which includes Sudakov safety) based on continuity properties on the space of events was proposed. In that context, $N$-(sub)jettiness [40,41] plays the role of a universal safe companion.

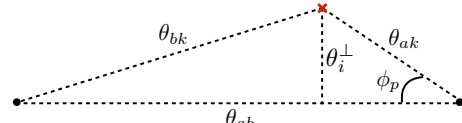

Figure 4: Illustration of the pull angle $\phi_{\mathrm{p}}$ and transverse angle $\theta_i^\perp$, defined by an emission denoted by the red cross. In this figure, we assume that the emission is closer to hard parton $a$ than hard parton $b$, $\theta_{ak} < \theta_{bk}$, which is why the pull angle is defined as the angle between the sides of the triangle of lengths $\theta_{ab}$ and $\theta_{ak}$.

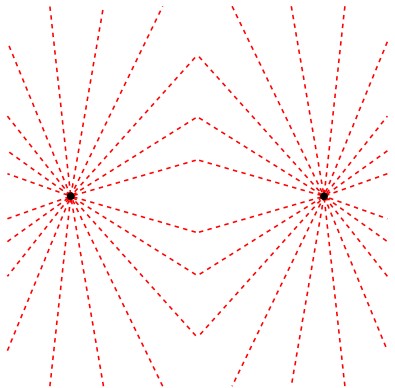

Figure 5: Contours of constant pull angle $\phi_{\mathrm{p}}$ with the location of the final-state hard partons denoted by the black dots.

joining the hard partons. This is illustrated in Fig. 4. In the collinear limit of the hard partons, the pull angle $\phi_{\mathrm{p}}$ is related to the pairwise angles between the hard partons and the emission by the law of cosines:

$$\max[\theta_{ak}^2, \theta_{bk}^2] = \min[\theta_{ak}^2, \theta_{bk}^2] + \theta_{ab}^2 - 2\theta_{ab} \min[\theta_{ak}, \theta_{bk}] \cos\phi_{\mathrm{p}}. \tag{15}$$

Solving for $\cos\phi_{\mathrm{p}}$ we have

$$\cos\phi_{\mathrm{p}} = \frac{\min[\theta_{ak}^2, \theta_{bk}^2] + \theta_{ab}^2 - \max[\theta_{ak}^2, \theta_{bk}^2]}{2\theta_{ab} \min[\theta_{ak}, \theta_{bk}]}. \tag{16}$$

We note that this is not related to the jet color ring by a monotonic function; therefore, the pull angle $\phi_{\mathrm{p}}$ is necessarily a worse color-representation discrimination observable than $\mathcal{O}$, given the assumptions we have made. Fig. 5 shows contours of fixed values of the pull angle $\phi_{\mathrm{p}}$, demonstrating that these contours are not identical to that of $\mathcal{O}$.

## 2.2 Relationship to Dipolarity

Unlike the pull angle, dipolarity $\mathcal{D}$ [2] is an IRC safe observable, and so is weighted both by relative angles as well as particle energies. It is defined as

$$\mathcal{D} = \frac{1}{\theta_{ab}^2} \sum_{i \in J} z_i (\theta_i^\perp)^2, \tag{17}$$

where the sum runs over all particles in a larger jet $J$ that contains the ends of the dipole separated by $\theta_{ab}$. $z_i$ is the transverse momentum or energy fraction of particle $i$ in the jet,

and $\theta_i^{\perp}$ is the transverse angle to the line connecting the ends of the dipole, as illustrated in Fig. 4. The dipolarity is also closely related to projections of the pull vector introduced in Ref. [12]. As dipolarity is an IRC safe observable, it is necessarily not monotonically related to the jet color ring, and therefore is a worse color discriminant under the assumptions we have made. Because of the energy weighting, the value of dipolarity does not uniquely determine the location of the dominant emission with respect to the ends of the dipole.

Nevertheless, it is still interesting to relate the angular dependence of dipolarity to the jet color ring. From Fig. 4 and application of Heron's formula, we find that the transverse distance $\theta_i$ of the emission off of the line separating the ends of the dipole is

$$
\begin{aligned}
(\theta_i^{\perp})^2 &= \frac{\theta_{ak}^2 + \theta_{bk}^2}{2} - \frac{(\theta_{ak}^2 - \theta_{bk}^2)^2}{4\theta_{ab}^2} - \frac{\theta_{ab}^2}{4} \\
&= \frac{\theta_{ab}^2}{2}\mathcal{O} - \frac{(\theta_{ak}^2 - \theta_{bk}^2)^2}{4\theta_{ab}^2} - \frac{\theta_{ab}^2}{4},
\end{aligned}
\tag{18}
$$

where in the second line we have inserted the expression for $\mathcal{O}$ from Eq. (13). This transverse angle is not monotonically related to the jet color ring, so even just its measurement is a worse color-representation discriminant than $\mathcal{O}$.

# 3 Distributions in Simulation

We now study the jet color ring in Monte Carlo simulations and compare its performance against the aforementioned pull angle and dipolarity, in order to test the robustness of the one-gluon-emission approximation to build the likelihood ratio. We will also consider the observable $D_2$ [26], a common boosted-boson tagger that aims to distinguish two-pronged jets from QCD background jets, which are mostly one-pronged. For our simulations we employ MAD-GRAPH5_AMC@NLO 2.6.7 in LO mode [42], with parton showers and non-perturbative effects supplied by PYTHIA 8.244 [43, 44]. 1M events were produced for each signal and background sample. The event analysis is implemented as a routine for the RIVET 3 analysis toolkit [45]. We consider three different analysis scenarios for QCD-singlet resonances via boosted hadronic decay reconstruction: $H \to b\bar{b}$ decay, $H \to gg$ decay, and $Z \to q\bar{q}$ decay. In all cases, we associate these resonances with a $Z \to \mu\mu$ decay, providing a massive recoil partner for the boosted kinematics, as as well as a clean leptonic trigger signature and estimator of the typical energy scale of the system.

### Analysis 1: $H \to b\bar{b}$

In the first analysis we want to distinguish a boosted Higgs decaying into a pair of bottom quarks from the $b\bar{b}$ background, mostly given by $g \to b\bar{b}$. Thus, we generate signal events from the process $pp \to Z(\mu\mu) + H(b\bar{b})$, and the background using the $pp \to Z + b\bar{b}$ process. The $b\bar{b}$ pair is required to be generated at matrix-element level for efficiency, thus excluding possible production at lower scales in the parton cascade or from secondary partonic interactions: the rate of such production has been verified to be negligibly small. Both matrix elements are enhanced by MLM merging [46] with the equivalent processes plus one additional final-state parton, to enable better modelling of the jet kinematics and possibly the boosted jet structure.

For each event, we reconstruct jets with the anti-$k_t$ algorithm [47] with radius $R = 1.0$, using all particles within $|\eta| < 5$ except the muons from the hard-scattering, and demand jet $p_T > 250$ GeV for consistency with the $p_T > 2M$ heuristic for the boosted-configuration threshold of a hadronically decaying particle of mass $M$. The generation acceptance for this

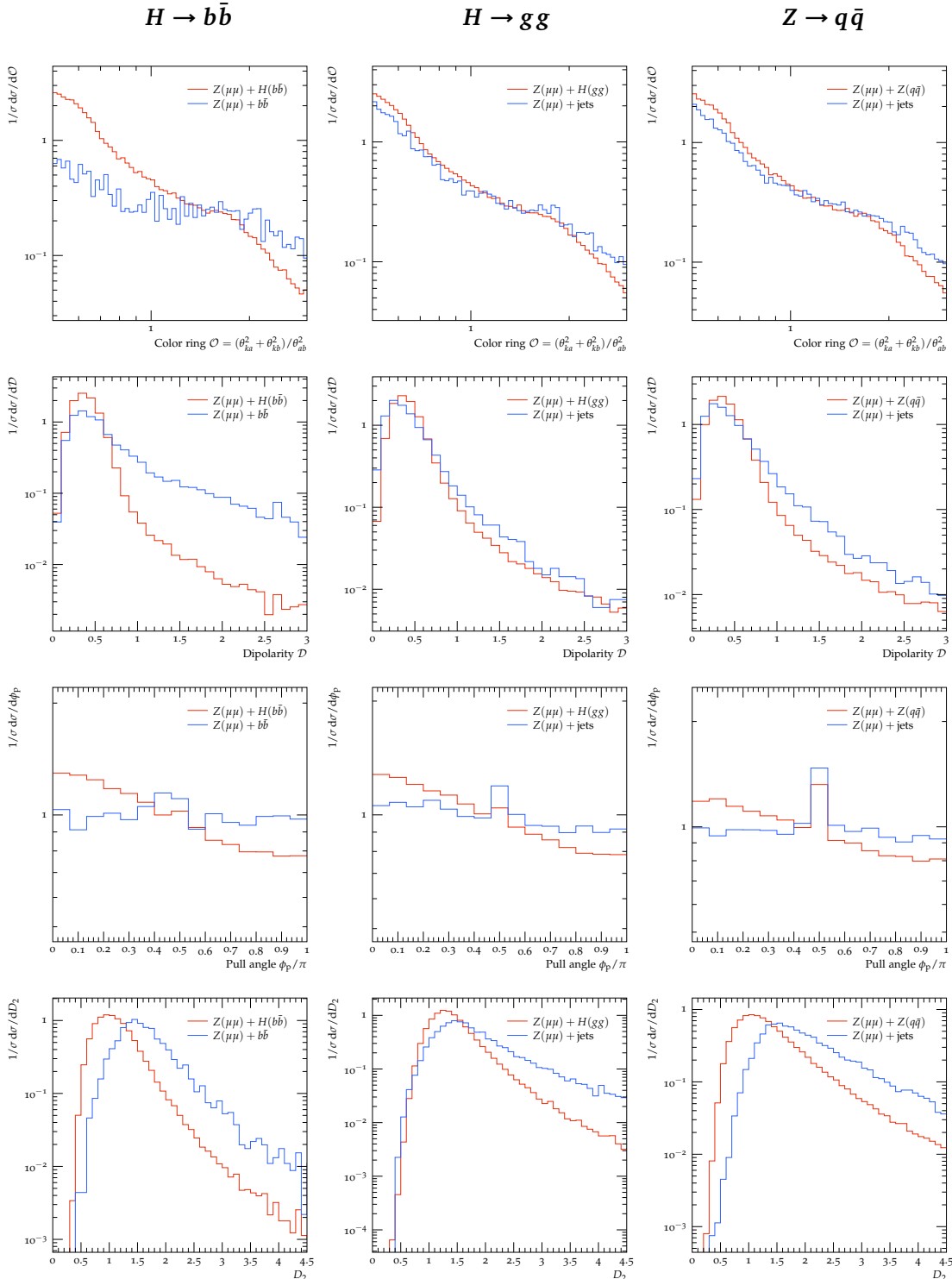

Figure 6: Large-$R$ jet structure distributions for $Z(\mu\mu)H(b\bar{b})$ vs. $Z(\mu\mu)b\bar{b}$ (left column), $Z(\mu\mu)H(gg)$ vs. $Z(\mu\mu)j(j)$ (central column), and $Z(\mu\mu)Z(q\bar{q})$ vs. $Z(\mu\mu)j(j)$ (right column). The observables, from top to bottom rows, are the jet color ring $\mathcal{O}$, dipolarity $\mathcal{D}$, jet pull angle $\phi_{\mathrm{p}}$, and $D_2$.

cut was enhanced, without biasing within the analysis acceptance, by use of a $\hat{p}_T^{\mu\mu} > 200$ GeV cut at matrix element level. Prompt charged leptons with $p_T > 20$ GeV are identified, and jets discarded if any such leptons are found within the jet radius. At this point the highest-$p_T$

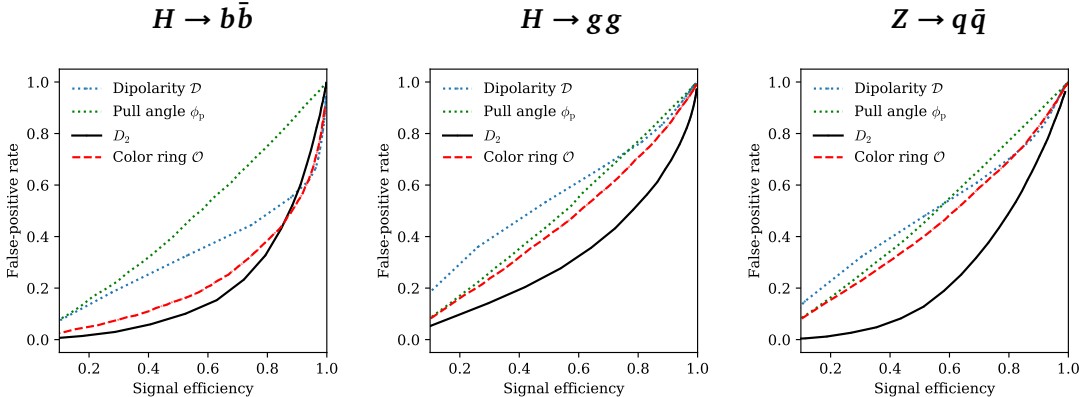

Figure 7: ROC curves for the $H \to b\bar{b}$, $H \to gg$, and $Z \to q\bar{q}$ analyses described in Section 3. These curves characterise the trade-off between the (desirable) true-positive event identification rate (aka efficiency), and the (undesirable) false-positive identification rate. Curves are shown for selections defined by sliding $x < C$ cuts, for all possible values of the cut $C$ and observables $x$ from the set of jet color ring $\mathcal{O}$, dipolarity $\mathcal{D}$, pull angle $\phi_\mathrm{p}$, and $D_2$.

jet is identified as the single jet of interest, and for this specific analysis is required to have a mass $m_J \in [110, 140]$ GeV $\sim m_H$ and (purely for computational efficiency) at least one ghost-associated [48] $b$-hadron with $p_T > 5$ GeV. These requirements enforce consistency with a boosted $H \to bb$ decay. In parallel with these large-$R$, all-particle jets, we identify "track-jets" with $p_T > 5$ GeV built from charged final-state particles with $p_T > 500$ MeV, using the anti-$k_t$ distance measure with a small $R = 0.2$ clustering radius: the restriction to charged particles reflects the higher experimental angular resolutions achievable with tracking as opposed to calorimetry. These are reduced to the smaller set of track-subjets found within $\Delta R < 0.8$ of the large-$R$ jet centroid, which are categorised as $b$-subjets if the track-jet has $p_T > 10$ GeV and a ghost-associated $b$-hadron with $p_T > 5$ GeV, or light subjets if they do not. For the $H \to bb$ analysis, at least two $b$-subjets are required, and the pair with the highest combined invariant mass are considered to define the ends of the color-ring dipole[3]. A third subjet, if present, is used as a proxy for the extra emission in the color ring variable — we note the loss of efficiency due to this third-subjet requirement. The angles appearing in the definition of the jet color ring, Eq. (13) are defined as the distances between the appropriate subjets, in the azimuth-pseudorapidity plane.

The resulting unit-normalised $H \to b\bar{b}$ observable distributions, for signal and background respectively, are shown in the leftmost plots of Fig. 6. The first row shows the color ring observable in which, as expected, the signal sample mostly populates the $\mathcal{O} < 1$ region. This gives rise to a prominent peak at small $\mathcal{O}$, while the background distribution is much flatter: it is also maximal at small values but in this unit-normalised form the background "peak" is a factor of $\sim 4$ lower than the signal. By comparison, the second, third, and final rows show the dipolarity $\mathcal{D}$, the pull angle $\phi_\mathrm{p}$,[4] and $D_2$ distributions respectively. All three also peak more at small values for signal than background, with less-obviously pronounced signal peaking than for $\mathcal{O}$: the dipolarity peaks in the same location $\mathcal{D} \sim 0.3$ for both signal and background, but with the signal exceeding the background by a factor of only $\sim 1.5$; the signal pull-angle

---

[3]In practice there is little ambiguity, as there are rarely more than two $b$-tagged subjets. Choosing the pair with highest individual $p_T$ values has also been investigated and gives near identical results for this analysis, and larger but not transformative differences for Analyses 2 and 3 in which there is no $b$-tag requirement.

[4]Specifically the "backward" pull angle, defined as the acute angle between the track-subjet dipole and the pull-vector computed from the constituents of the subleading-$p_T$ track-subjet.

density falls monotonically from small to large angles as compared to a flat background, but its peak only exceeds that of the background by 30%; and $D_2$ peaks to the same maximum density for both signal and background, but with a clearer separation in their peak location than for $\mathcal{O}$ and $\mathcal{D}$. It therefore appears that the color ring $\mathcal{O}$ does indeed have a signal/background separation power equal to or greater than other observables designed with this application in mind.

## Analysis 2: $H \rightarrow gg$

In the second analysis, we instead concentrate on the rather challenging decay (via intermediate heavy-quark loops) of a Higgs boson into gluons. The setup for the signal simulation, as well as the kinematical cuts, are the same as in the previous analysis. However, we cannot rely on $b$-quarks in the final state and so the dominant background is general $Z$ production in association with jets, which is dominated by light-quark and gluon jets. Our signal process is hence a merged sample from $pp \rightarrow Z(\mu\mu) + H(gg) + \leq 1j$ matrix elements, and the Born background process is $pp \rightarrow Z(\mu\mu) + j + \leq 2j$, where the $j$ in all cases represents any color-charged parton. The merging of the background sample with up to *two* extra parton emissions reflects that the Born process only contains a single jet, an inclusive approach that allows production of the two or three prong jet structure either by matrix-element or by parton-shower emissions.

The analysis procedure follows that of the $H \rightarrow b\bar{b}$ above, except for the $b$-tag requirements on both the large-$R$ jet and the small-$R$ track-subjets: all track-jets with $p_T > 5$ GeV are considered viable leading subjets. The distributions for this analysis are shown in the middle column of Fig. 6. We see again that the signal distribution behaves as expected from our theoretical analysis, which was based on essentially a one-gluon emission approximation in the soft limit, i.e. it is sharply peaked at small $\mathcal{O}$. However, contrary to our naive expectation, the background is now also strongly peaked with a similar shape. This is mostly likely due to a breakdown of our factorized "production×decay" approximation, which for the background heavily relies on the collinear limit of the tagged subjets. The color ring still isolates a larger fraction of the signal than the background into the $\mathcal{O} < 1$ region, but the separation power is clearly reduced. Similar changes to the background topology are also seen in the dipolarity $\mathcal{D}$, which becomes nearly identical for signal and background, and to a lesser extent in the pull angle and $D_2$ distributions.

## Analysis 3: $Z \rightarrow q\bar{q}$

In our third analysis, we look at double $Z$ production, with one $Z$ decaying into muons and one hadronically, in the equivalent boosted regime $p_T > 180$ GeV $\sim 2m_Z$ so that the hadronic decay products of the $Z$ are reconstructed as a single (two-pronged) jet. Equivalent to the $H \rightarrow gg$ analysis, our signal sample is the merged $pp \rightarrow Z(\mu\mu) + Z(q\bar{q}) + \leq 1j$ process, and the background $pp \rightarrow Z(\mu\mu) + j + \leq 2j$. The analysis is similarly as for $H \rightarrow gg$, with no $b$-tagging requirements, but with the leading jet $p_T$ cut reduced as motivated above, and the jet-mass window shifted to $m_J \in [75, 105]$ GeV for compatibility with the $Z$ pole mass. The resulting distributions are again shown in Fig. 6, in the rightmost column. Because the background is the same as the one we considered in our previous analysis, albeit with slightly different kinematical cuts, the background distribution still presents the unwanted peak at small $\mathcal{O}$, while the signal distribution has the shape we expect from a singlet decay. Sightly more signal/background separation is present for this lower-scale analysis and signal process.

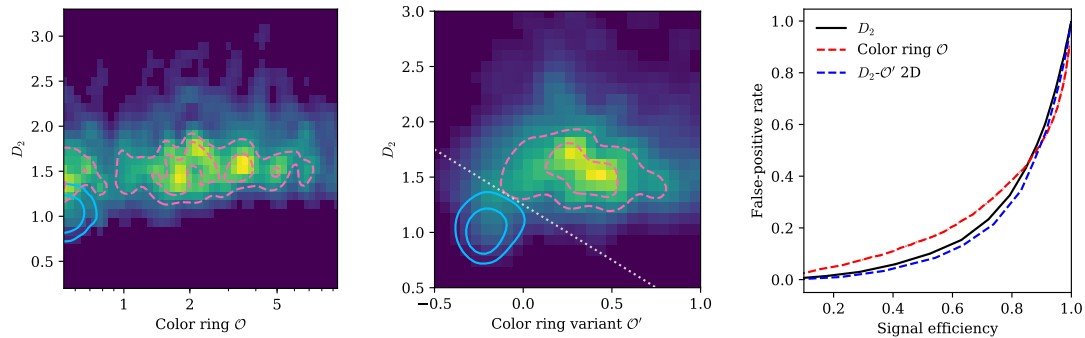

Figure 8: Illustrations of complementarity of $D_2$ with the color ring observables. The first two figures show the 2D distributions of $D_2$ with the color ring $\mathcal{O}$ and its $\mathcal{O}'$ variant, with contour overlays at 50% and 75% of the maximum values for signal (solid) and background (dashed) densities separately, to indicate their main concentrations. Some orthogonality is seen between the observables, suggesting additional separation power from a 2D cut. The third plot shows the previous ROC curves for $D_2$ and $\mathcal{O}$, compared to the performance of a simple 2D $D_2 + \mathcal{O}' < C$ cut, an example of which is indicated by a dotted line in the middle plot. Mild performance improvements over $D_2$ alone are seen for all values of signal efficiency.

## Interpretation

By looking at the three analyses together, we can draw two important lessons. First, the jet color ring does behave as expected on signal jets which are originated by the boosted decay of a color-singlet resonance, and the resulting distributions depend only mildly on the spin of the decaying particle, making it a rather universal color-singlet tagger. However, we can already anticipate by looking at the background distributions that the jet color ring is an efficient tagger only when the task is to distinguish between two well-defined color configurations, as is the case $H \to b\bar{b}$ (color singlet) vs $g \to b\bar{b}$ (color octet). Unfortunately, when the background is characterized by more complex color configurations, as in the QCD jet case, we anticipate the discrimination power to be rather poor. In this sense the $H \to b\bar{b}$ (and, although not explicitly considered here, $Z \to b\bar{b}$) processes are experimentally privileged with respect to other hadronic substructure channels by the power of the $b$-tagging to select out the quark–antiquark dipole, while flavor-inclusive analyses can be confused by presence of multiple types of dipole constituents, because $q$ and/or $\bar{q}$ do not necessarily correspond to the leading subjets.

We can make these considerations more quantitative by looking at ROC curves corresponding to $x < C$ one-sided cuts, for variable $x$ chosen from the variables considered here: this cut is appropriate as smaller values are more signal-like for all three distributions. A ROC curve is a plot of the true-positive rate (the fraction of signal events identified by the reconstruction analysis and the selection cut) against the false-positive rate (the fraction of background events selected), and captures — in a form independent of the cut values for the different variables — the trade-off between the acceptance and purity of the signal-identification strategy. Sets of such curves are computed for each of the three analyses from the distributions already considered, and are shown in Fig. 7. As anticipated, the performance of the jet color ring is good in the first ($H \to b\bar{b}$) case but is rather poor when it comes to discriminating against QCD-jet backgrounds.

In the plots of Fig. 7 we also show the ROC curve corresponding to the two-prong tagger $D_2$. This jet shape is often used as a vector boson tagger and therefore it comes without surprise that it performs well in that context. Its discrimination power is also good in the case of gluonic

Higgs decay, which features a similar topology. Surprisingly, $D_2$ also outperforms our color-ring tagger in the $H \to b\bar{b}$ analysis for medium to low efficiencies. This result is somewhat counter-intuitive because both signal and background feature a two-prong structure: $D_2$ must be exploiting different information to discriminate the two. We note that $D_2$ is a jet shape that takes as input all particles in a jet and it is therefore sensitive to soft radiation, the pattern of which is dictated by the color state of the emitting dipole. This observation motivates further studies to analytically investigate the sensitivity of $D_2$ to soft radiation in more detail than in its original publication [26].

As $D_2$ and $\mathcal{O}$ both probe the color configuration of the decaying particle, but with different strategies, it is natural to try and combine them. In Fig. 8, we show the total expected event populations (with background and signal appropriately normalised) for the $H \to b\bar{b}$ analysis in the planes of $D_2$–$\mathcal{O}$ and $D_2$–$\mathcal{O}'$, where $\mathcal{O}'$ is a monotonic variant (for fixed kinematic of the two tagged subjets $a$ and $b$) of the color ring previously considered, defined as

$$\mathcal{O}' = \theta_{ak}^2 + \theta_{bk}^2 - \theta_{ab}^2 \,. \tag{19}$$

This has been introduced as, unlike $\mathcal{O}$, its signal peak is empirically seen to be separated from the background one, making it a more obvious candidate for a 2D background separation cut. This feature, however, comes at the cost of losing the desirable boost-invariance property of $\mathcal{O}$.

From the width of the distributions in the middle plot of Fig. 8, it is clear that the two variables are not just reparametrisations of one another but contain significant orthogonal information. The 1D cuts previously discussed correspond to vertical ($\mathcal{O}'$) or horizontal ($D_2$) cut lines, but it is evident that such cuts also include background contamination which is well isolated in the other variable. Hence, greater background rejection for any signal efficiency is possible by use of a two-dimensional cut in the plane. We do not attempt to generally identify an optimal shape for this cut, but in the rightmost figure we compute a ROC curve for a simple linear $D_2 + \mathcal{O}' < C$ cut, i.e. a straight-line cut with negative unit gradient in the $D_2$–$\mathcal{O}'$ plane. This cut performs marginally better than either $D_2$ or $\mathcal{O}'$ alone, suggesting that despite the unexpectedly high selection power of $D_2$ in this context, additional information such as the jet structure relative to the experimentally identified $b$–$\bar{b}$ dipole can add power to $b\bar{b}$ singlet-resonance search analyses.

# 4 Conclusion and Outlook

In this study we have introduced a new observable, the jet color ring, that is sensitive to the color configuration of a decaying particle. While tagging observables are always defined with the idea of disentangling signal and background by exploiting their radiation pattern, our approach is, to the best of our knowledge, novel because it attempts to build a simple color-singlet tagger directly from the QCD likelihood ratio in a certain kinematic limit. We have demonstrated through Monte Carlo simulations that the jet coloring exhibits good discrimination power in the context of the Higgs boson decay into bottom quarks and offers complementary information to more sophisticated taggers based on jet shapes, such as $D_2$. The simulations also show that tagger performance is worsened when one has to deal with backgrounds which have richer color structure, as in the case of the Higgs, or vector boson, decay into light jets. Given the relative simplicity of the color ring observable, it would be interesting to further study its performance from a theoretical point of view, looking at, for instance, its perturbative behavior and its dependence on non-perturbative physics. We leave these studies for future work.

# Acknowledgments

We thank the organizers of the QCD@LHC 2019 workshop in Buffalo NY where this project was started. We thank Michael Spannowsky for reading the draft of this paper. The work of SM is supported by Università di Genova under the curiosity-driven grant "Using jets to challenge the Standard Model of particle physics" and by the Italian Ministry of Research (MIUR) under grant PRIN 20172LNEEZ. AB is supported by Royal Society fellowship grant UF160548, and acknowledges research funding from the European Union's Horizon 2020 research and innovation programme under the Marie Skłodowska-Curie Action Innovative Training Networks MCnetITN3 (grant agreement No. 722104). AB and GC have received funding from the UK Science and Technology Facilities Council (STFC) particle physics consolidated grant programme.

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
