# Peer review of "An Optimal Observable for Color Singlet Identification"

_SciPost Physics, doi:SciPost Phys. 9, 026 (2020)_

## Round 1 · Referee Report · Anonymous · 2020-7-11

Strengths

1- Introduces a jet substructure observable based directly on the likelihood ratio of signal vs. background jets. This seems to be the first study that does so, though there are antecedents in the shower deconstruction literature.

2- Demonstrates the method convincingly with Monte Carlo event samples and performs an apples-to-apples comparison to previous observables.

3- Will likely inspire future developments in jet physics based on similar principles.

4- Overall well motivated and clearly written.

Weaknesses

1- Some of the notation is not fully explained (see requested changes below).

2- There is a confusing step in the derivation that might be related to the suboptimal performance seen in two of the case studies (see requested changes below).

Report

This manuscript introduces a new jet substructure observable called the "jet color ring", designed to identify boosted color singlet resonances with hadronic decays. The key innovation is defining the observable via the likelihood ratio of signal vs. background jets, which is provably optimal in the appropriate limit and under reasonable assumptions. This likelihood ratio is estimated using leading order matrix elements in the soft emission limit. Using Monte Carlo event samples, the authors demonstrate improved tagging performance relative to previous observables in the literature designed for the same task, namely jet pull and dipolarity. The improved performance is most noticeable for the case of a Higgs boson decaying to a bottom quark pair, which is an important target for the LHC.

To my knowledge, this is the first paper to define a jet substructure observable in this way, though as mentioned by the authors in footnote 1, a similar likelihood ratio philosophy is used in the shower deconstruction literature. For this reason, I support eventual publication of this paper in SciPost Physics. That said, I noticed a number of (relatively small) issues while reading the manuscript, detailed below, which I would request that the authors address. The large number of suggestions should not be viewed as a negative opinion about the manuscript, but rather as reflecting my interest and excitement that methods like this could be a powerful strategy for future jet substructure innovations. I hope the authors find these suggestions valuable.

Requested changes

1- First, there is a delicate issue related to the title. The word "color" has a technical meaning in QCD, of course, but I think it would be advisable to avoid the phrase "color discrimination" in the title. This paper originally appeared on the arXiv on Juneteenth, which is an important day in U.S. history, though perhaps not as well known to an international audience. While I don't think the authors intended to reference the discriminatory past (and present) treatment of people of color, the association might be automatic in some readers' minds. Further, as a physics point, the authors' technique is really for identifying color singlet objects, since their observable would not be appropriate for quark/gluon discrimination (which is what I thought this paper would be about from just looking at the title). For these reasons, I would suggest that the authors consider a title more along the lines of "An Optimal Observable for Color Singlet Identification", or something similar.

2- Related to point 1, change "color tagger" to "color singlet tagger" in the abstract.

3- In the second paragraph of the introduction, the authors reference electroweak boson decays to "QCD jets". I would change this to "light quark jets", since QCD jets could refer to both quark and gluon jets (and is used in that way later in the manuscript).

4- (See weakness 1 above.) In equation (1) and the surrounding text, the color matrices are not clearly defined. To make this paper more stand alone, I think the bold-faced $\mathbf{T}$ notation should be defined explicitly, and perhaps more references to the literature given that use this color matrix notation. Related to this, I think the authors should clarify what the color indices $\alpha$ and $\beta$ refer to, i.e. whether they label overall representations (triplet, octet) or do they label the states ($\alpha \in [1,2,3]$ for a color triplet). This is relevant to understand exactly what the sum in equation (6) is referring to.

5- In equation (3), it might help to have a bold-faced $\mathbf{1}$ in the last step to refer to the identity matrix.

6- (See weakness 2 above.) I am confused about the second step in equation (6), where the authors swap $\beta$ and $\alpha$ under the sum. In the case that the outgoing partons have different color representations (i.e. quark to quark + gluon), this derivation doesn't seem to be correct on the face of it, though this is related to the notational question in point 4 about what the matrices and indices are referring to.

7- The authors might want cite the literature for the derivation of equation (7), or at least say that this equation comes from rearranging and squaring equation (5).

8- I found Figure 1 to be very helpful. The authors should consider enhancing this figure by adding the diagram for color singlet decays as well, to make it easier to compare the signal and background processes.

9- Below equation (9), the authors refer to a background from "all $1 \to 2$ QCD splittings". Because the background is a mixture of quark and gluon jets, I don't see how this derivation can be independent of the quark/gluon fraction of the background sample.

10- I found Figure 2 to be very helpful. The authors should consider enhancing this figure by adding more contour lines (so that it is obvious that all of the contours are just circles). In addition, the authors should consider adding contour plots for the pull angle and dipolarity to show more viscerally why those observables have suboptimal structure.

11- Below equation (13), the authors emphasize that the jet color ring is IRC unsafe. Given (two of) the authors, I am sure they have thought about whether the jet color ring is Sudakov safe. Could they comment on that, particularly in relation to their previous work on the Sudakov-safe pull angle?

12- Also below equation (13), the authors say that they will give a "precise particle-level definition" later. They might consider adding "in terms of subjet kinematics" as a quick preview of the method to come.

13- I found Figure 3 to be very helpful, but the notation $\theta_i$ threw me for a loop, since that notation often refers to the angle of a particle to the jet axis. To emphasize that this is a transverse angle, maybe the authors should consider $\theta^\perp_i$ instead?

14- In the preprocessing step for the $H \to b \bar{b}$ analysis, the authors require at least one ghost-associated $b$-hadron. Why only one and not two? (I thought it might be because the same preprocessing steps would be used later, but the $H \to gg$ analysis does not require a $b$-tag.) A requirement of two $b$-subjets is used later, so I don't think there is a mistake; perhaps the one $b$-hadron requirement is just a remnant from a study not shown in the text?

15- (See weakness 2 above.) In the central and right columns of Figure 4, the background peaks more strongly than expected from the argument in Section 2. The authors point this out by saying that this behavior is "contrary to our naive expectation", but I think more discussion of this is needed. First, it is possible (though not likely) that there is an issue with equation (6), related to point 6 above, such that the jet color ring is not actually the optimal observable when the background is dominated by quark/gluon jets. Second, it is possible that focusing on soft (instead of collinear) physics is the reason for the poor performance. Third, it is possible that the authors should have selected the pair of subjets that form a dipole with the highest invariant mass (instead of the two highest $p_T$ subjets) to define their observable. There may be other possibilities as well. But given the large discrepancy in the behavior compared to expectations, I think a bit more discussion is needed here, since otherwise it would seem to call into question the validity of the likelihood ratio philosophy.

16- (See weakness 2 above.) Related to the point above, the authors blame the poor performance on "more complex color configurations". If the derivation in Section 2 is to be believed, in particular the consideration of all $1 \to 2$ QCD splittings (as mentioned in point 9 above), then the jet color ring should still be the optimal observable in the soft limit even for mixed samples of quark/gluon jets. So I expect the authors will want to revisit this wording depending on how they respond to point 15 above, particularly if it turns out that it is the more complicated collinear structure that explains the poor performance.

17- The authors refer to equation (19) as a "monotonic variant" of the color ring. Previously, around equation (10), the authors use monotonic to refer to changes to the observable that do not change the discrimination power. In that sense, equation (19) is not a monotonic variant, since $\theta_{ab}$ is different for different jets, so the discrimination power is different (as clearly seen in the left and middle panels of Figure 6). So I think an alternative language should be used here.

  • validity: high
  • significance: high
  • originality: high
  • clarity: high
  • formatting: excellent
  • grammar: excellent

Author:  Andy Buckley  on 2020-07-24  [id 908]

(in reply to Report 1 on 2020-07-11)
Category:
answer to question
reply to objection

Our thanks for the very constructive report. We have addressed the requested changes as described point-by-point below, and consider the paper much improved by the feedback. We will resubmit with a new manuscript via arXiv if the responses below are considered sufficient.

1 - Changed

2 - Changed

3 - Changed

4 - Added explanation and references

5 - Changed

6 - Added after equation (6) "where we have exploited the fact that our analysis is blind to both the color and the flavor of the final-state partons."

7 - Changed

8 - Inserted the color singlet figure now. Updated the text to mention the new figure.

9 - We acknowledge that our notation was indeed confusing. Eq.(1) and Eq.(2) do not represent the full matrix element square for signal and background, respectively. Rather, we have exploited the collinear limit of the two subjets we are tagging and factorized their contribution from the complete expression. The remaining part of the matrix element squares, which include both initial-state contributions such as parton distribution functions, as well as eventual extra jets, contributes with an overall factor which is not relevant, within our approximation, for the construction of the observable. We have changed the sentence just before Eq.(1) to be: "The matrix element square describing the decay of a boosted color-singlet state into two hard partons plus a single low-energy gluon is". Furthermore, we have rewritten the paragraph before Eq.(8).

10 - Added more lines and a contour plot for pull. Updated the caption and in-text paragraph describing figure 3 appropriately. Added a reference to the new pull angle contours plot in the paragraph about relationship to the pull angle.

11 - Added "Such jet shape may be Sudakov safe~\cite{Larkoski:2013paa,Larkoski:2014wba,Larkoski:2015lea}, although it is not clear what its natural safe companion observable would be~\footnote{In a recent study~\cite{Komiske:2020qhg}, a generalized defintion of IRC safety (which includes Sudakov safety) based on continuity properties on the space of events was proposed. In that context, $N$-(sub)jettiness~\cite{Stewart:2010tn,Thaler:2010tr} plays the role of a universal safe companion.}"

12 - Changed

13 - Changed

14 - This is simply a description of the algorithm as implemented and attached to the arXiv submission, which makes a minimal flavour-tag requirement on the large-R jet before the relatively expensive computation of matched, b-tagged subjets used to apply a tighter requirement. The "preselection" role of this cut has now been clarified in the text.

15 - The first two comments made here are addressed below in point 16. Addressing the 3rd comment, we appreciate the suggestion for alternative dipole-subjet selection via maximum invariant mass. We have implemented and tested this alternative scheme, and no change is seen for the bb subjet selection (unsurprisingly, as there is rarely a 3rd tagged subjet to give ambiguity), but more substantial changes are seen for the H->gg and Z->qq analyses. These are not significant enough to change the thrust of the results, but give a slight improvement in ROC performance for dipole-based measures in H->gg, and a more ambiguous shift of the ROC curves for Z->qq: we have hence chosen to make the max-mass scheme our default and updated the paper and plots accordingly, with a note about the impact of using an alternative leading-pTs scheme.

16 - We believe that the additional explanation added in order to address the Referee's point 9) partly answers this objection too. Furthermore, we have modified the sentence as follows: "However, contrary to our naive expectation, the background is now also strongly peaked with a similar shape. This is mostly likely due to a breakdown of our factorized "production$\times$decay" approximation, which for the background heavily relies on the collinear limit of the tagged subjets. "

17 - We have clarified that the observable O' is a monotonic variant of the jet color ring for "fixed kinematic of the two tagged subjets a and b."

---

## Round 2 · Referee Report · Anonymous · 2020-8-14

Report

The authors have addressed all of my previous concerns and I am happy to recommend publication in SciPost Physics.

---

## Round 2 · Author Response

This is the improved version, taking the reviewer feedback into account. Our thanks to the reviewers, whose comments were both complimentary and helpful.

---

## Round 2 · List of Changes

We have addressed the requested changes as follows:

1 - Changed

2 - Changed

3 - Changed

4 - Added explanation and references

5 - Changed

6 - Added after equation (6) "where we have exploited the fact that our analysis is blind to both the color and the flavor of the final-state partons."

7 - Changed

8 - Inserted the color singlet figure now. Updated the text to mention the new figure.

9 - We acknowledge that our notation was indeed confusing. Eq.(1) and Eq.(2) do not represent the full matrix element square for signal and background, respectively. Rather, we have exploited the collinear limit of the two subjets we are tagging and factorized their contribution from the complete expression. The remaining part of the matrix element squares, which include both initial-state contributions such as parton distribution functions, as well as eventual extra jets, contributes with an overall factor which is not relevant, within our approximation, for the construction of the observable. We have changed the sentence just before Eq.(1) to be:
"The matrix element square describing the decay of a boosted color-singlet state into two hard partons plus a single low-energy gluon is".
Furthermore, we have rewritten the paragraph before Eq.(8).

10 - Added more lines and a contour plot for pull. Updated the caption and in-text paragraph describing figure 3 appropriately. Added a reference to the new pull angle contours plot in the paragraph about relationship to the pull angle.

11 - Added "Such jet shape may be Sudakov safe~\cite{Larkoski:2013paa,Larkoski:2014wba,Larkoski:2015lea}, although it is not clear what its natural safe companion observable would be~\footnote{In a recent study~\cite{Komiske:2020qhg}, a generalized defintion of IRC safety (which includes Sudakov safety) based on continuity properties on the space of events was proposed. In that context, $N$-(sub)jettiness~\cite{Stewart:2010tn,Thaler:2010tr} plays the role of a universal safe companion.}"

12 - Changed

13 - Changed

14 - This is simply a description of the algorithm as implemented and attached to the arXiv submission, which makes a minimal flavour-tag requirement on the large-R jet before the relatively expensive computation of matched, b-tagged subjets used to apply a tighter requirement. The "preselection" role of this cut has now been clarified in the text.

15 - The first two comments made here are addressed below in point 16. Addressing the 3rd comment, we appreciate the suggestion for alternative dipole-subjet selection via maximum invariant mass. We have implemented and tested this alternative scheme, and no change is seen for the bb subjet selection (unsurprisingly, as there is rarely a 3rd tagged subjet to give ambiguity), but more substantial changes are seen for the H->gg and Z->qq analyses. These are not significant enough to change the thrust of the results, but give a slight improvement in ROC performance for dipole-based measures in H->gg, and a more ambiguous shift of the ROC curves for Z->qq: we have hence chosen to make the max-mass scheme our default and updated the paper and plots accordingly, with a note about the impact of using an alternative leading-pTs scheme.

16 - We believe that the additional explanation added in order to address the Referee's point 9) partly answers this objection too. Furthermore, we have modified the sentence as follows:
"However, contrary to our naive expectation, the background is now also strongly peaked with a similar shape. This is mostly likely due to a breakdown of our factorized "production$\times$decay" approximation, which for the background heavily relies on the collinear limit of the tagged subjets. "

17 - We have clarified that the observable O' is a monotonic variant of the jet color ring for "fixed kinematic of the two tagged subjets a and b."

You are currently on this page

Resubmission 2006.10480v2 on 12 August 2020

---

## Editorial Decision

published